# Flavanonol Glycosides from the Stems of *Myrsine seguinii* and Their Neuroprotective Activities

**DOI:** 10.3390/ph14090911

**Published:** 2021-09-09

**Authors:** Hee-Ju Lee, Eun-Jin Park, Ba-Wool Lee, Hyo-Moon Cho, Thi-Linh-Giang Pham, Quynh-Hoa Hoang, Cheol-Ho Pan, Won-Keun Oh

**Affiliations:** 1Korea Bioactive Natural Material Bank, Research Institute of Pharmaceutical Sciences, College of Pharmacy, Seoul National University, Seoul 08826, Korea; hjlee81@kist.re.kr (H.-J.L.); eunjin_p@snu.ac.kr (E.-J.P.); paul36@snu.ac.kr (B.-W.L.); chgyand@naver.com (H.-M.C.); 2Natural Product Informatics Research Center, Korea Institute of Science and Technology, Gangeung 25451, Korea; panc@kist.re.kr; 3Department of Botany, Hanoi University of Pharmacy, Hanoi 100000, Vietnam; giangptl@hup.edu.vn (T.-L.-G.P.); hoahp@hup.edu.vn (Q.-H.H.)

**Keywords:** *Myrsine seguinii*, Primulaceae, Alzheimer’s disease, flavanonol glycoside, molecular networking, neuroprotective activity

## Abstract

The accumulation of amyloid beta (A*β*) peptides is common in the brains of patients with Alzheimer’s disease, who are characterized by neurological cognitive impairment. In the search for materials with inhibitory activity against the accumulation of the A*β* peptide, seven undescribed flavanonol glycosides (**1**–**7**) and five known compounds (**8**–**12**) were isolated from stems of *Myrsine seguinii* by HPLC-qTOF MS/MS-based molecular networking. Interestingly, this plant has been used as a folk medicine for the treatment of various inflammatory conditions. The chemical structures of the isolated compounds (**1**–**12**) were elucidated based on spectroscopic data, including 1D and 2D nuclear magnetic resonance (NMR), high-resolution electrospray ionization mass spectrometry (HRESIMS) and electronic circular dichroism (ECD) data. Compounds **2**, **6** and **7** showed neuroprotective activity against A*β*-induced cytotoxicity in A*β*_42_-transfected HT22 cells.

## 1. Introduction

Alzheimer’s disease (AD) is a neurodegenerative disorder characterized by progressive memory loss and behavioral abnormalities [1]. Due to the increase in the size of the elderly population, the number of patients with AD is currently estimated to be 5.5 million in the United States, and the number is expected to reach over 13.8 million by 2050 [2]. The main pathological features of AD are deposits of extracellular amyloid beta (A*β*) plaques. The insoluble A*β* protein and neurofibrillary tangles of the tau-protein, which are involved in microtubule maintenance in neural cells, ultimately result in memory dysfunction [3]. A*β* peptides composed of 39–42 amino acids are produced by the cleavage of amyloid precursor protein (APP) by *β*-secretase and *γ*-secretase [4]. In particular, A*β*_1-40_ and A*β*_1-42_, which are more neurotoxic than other A*β* peptides, are excessively produced in cerebral neural cells and accumulate in the form of senile plaques [5]. The accumulated senile plaques form protofibrils, fibrils and plaques by A*β* oligomerization [6], which has been implicated in neuronal dystrophy and synaptic loss through pathways such as neurotoxicity, oxidative stress and inflammatory reactions in both human and mouse models of AD [7,8,9]. The U. S. Food and Drug Administration (FDA) has approved drugs such as tacrine, donepezil, rivastigmine and galantamine (as acetylcholine esterase enzyme inhibitors) and memantine (as an N-methyl-d-aspartate (NMDA) receptor antagonist) for use in AD patients [10]. However, these drugs, which have side effects such as vomiting and hepatotoxicity, have no therapeutic effect and only cause temporary improvement [11]. Galantamine is a naturally occurring plant tertiary alkaloid that acts not only as a reversible inhibitor of acetylcholinesterase but also as an allosteric modulator of nicotinic receptors and is used as a treatment for mild-to-moderate Alzheimer’s disease [12]. Natural products are a good source of potential Alzheimer’s treatments with relatively few side effects and excellent efficacy. 

*Myrsine seguinii* H. Lév., belonging to the family Primulaceae, occurs as shrubs or trees that can grow up to 2–12 meters in height and is distributed in Southeast Asian countries such as Vietnam and China [13]. The plant mainly grows in tropical and subtropical regions of the world and has a limited distribution in the northern area of Japan [14]. The genus *Myrsine* comprises approximately 300 species of evergreen shrubs and trees, which have traditionally been used as a folk medicine to treat anti-inflammatory and infectious diseases in Southeast Asian countries [15,16]. *M. africana* is used as an anti-tapeworm infection agent and as a treatment for dropsy, colic and dysmenorrhea [17]. In particular, *M. seguinii* is used in Myanmar as a medicinal plant for treating colds, influenza and headaches, and it has been used to treat inflammatory diseases on Okinawa Island, Japan [18]. The phytochemical components of *M. seguinii* have been reported to include flavonol glycosides [19], anti-inflammatory prenylated benzoic acid derivatives and myrsinionosides consisting of hydroquinones and *p*-benzoquinones [20,21]. Inflammation has been reported as a distinctive feature of Alzheimer’s patients, along with two other pathological features (*β*-amyloid and neurofibrillary tangles) and the link between the initial accumulation of *β*-amyloid and the subsequent development of neurofibrillary tangles has been investigated [22]. *Myrsine seguinii* is a plant of interest for identifying potential AD therapeutic agents because it has traditionally been used to treat inflammation and is reported to contain chemicals with anti-inflammatory effects.

In this study, we attempted to investigate active compounds in a 70% EtOH extract of *M. seguinii* using the Global Natural Product Social Molecular Networking (GNPS) (https://gnps.ucsd.edu, accessed on 7 January 2019) open-access web-based platform, which forms molecular networks (MNs) [23]. Dereplication by high-performance liquid chromatography-quadrupole time-of-flight mass spectrometry (HPLC-qTOF MS/MS) resulted in the isolation of seven new flavanonol glycosides (**1**–**7**) along with five known compounds (**8**–**12**). In addition, the neuroprotective effects of the isolated compounds (**1**–**12**) were tested against the cytotoxicity induced in A*β*_42_-transfected HT22 cells.

## 2. Results

### 2.1. Authentication of M. seguinii H. Lév

In the case of plants used as traditional medicines, special attention should be paid to the authentication of samples, as their intake in the patients may cause severe side effects [24,25]. In the genus *Myrsine*, which consists of approximately 300 species of evergreen shrubs and trees, the authentication of *M. seguinii* was attempted based on plant morphology, cross-sectional observations and the DNA barcode technique. 

Morphologically, the studied samples were shrubs with a height of 3–4 m or trees with a height up to 10 m. The stems were highly branched and covered by cracked white bark. The branches were 3–5 mm in diameter and were white lenticellate, rugose, reddish puberulent and glabrescent in early stages. The leaves were simple and alternate, with petioles of 2–3 mm. The leaf blade was elliptic to narrowly linear-oblanceolate, 3–7 cm long, 1–2 cm wide, leathery, glabrous, base cuneate and reddish, with an entire margin and acute apex. Lateral veins were apparent, with 20–25 pairs (Figure 1A). The morphological features closely matched the description of *M. seguinii* in the Flora of China [26]. A cross-section of a leaf of *M. seguinii* is illustrated in Figure 1B. The upper and lower epidermis consisted of a single cell layer and covered the entire midrib surface. The upper and lower collenchyma was under the epidermal layer, with straight anticlinal wall cells. Calcium oxalate crystals and secretory schizogenous cavities were distributed in a scattered manner throughout collenchymal tissues. The vascular bundles occupied half of the transverse section area and were covered by 2–4 layers of sclerenchyma. The phloem consisted of 3–5 layers of small cells and covered the xylem. The order of the tissues from the lower to the upper parts of the leaf blade was as follows: lower epidermis, spongy mesophyll, palisade mesophyll and upper epidermis. The image of the stem cross-section shows the secondary structure of the stem of *M. seguinii* (Figure 1C). The outermost layer was the cork, which consisted of thick multiple cell layers. This layer was followed by multiple layers of parenchyma cells that showed a variety of shapes and could contain calcium oxalate crystals. The vascular bundles occupied more than half of the transverse section area and were arranged as cascade clusters. Each bundle consisted of secondary phloem, secondary xylem and primary xylem. The sclerenchyma cells covered the secondary phloem in the form of fiber bundles. The primary phloem was not visible. Several secretory schizogenous cavities could be observed inside the pith.

The DNA sequence was compared with published sequences available in GenBank (National Institutes of Health) using the basic Local Alignment Search Tool (BLAST) of the National Center for Biotechnology Information (NCBI) to generate the neighbor-joining (NJ) tree. The results of the DNA sequencing analysis and BLAST analysis showed high similarity with sequences from the genus *Myrsine* (Appendix A). The BLAST analysis of *trn*H-*psb*A sequences showed sequence similarity of 99% with *M. seguinii* distributed in Vietnam, Cambodia and China, and *M. umbellate* distributed in Ecuador and Brazil. The BLAST analysis of nrITS revealed 98% agreement with *M. segunii* and *M. chatthamica* from the Chatham Islands. The experimental sample was grouped with the genus *Myrsine* in the NJ tree based on *mat*K, *rbc*L, *trn*L-*trn*F and *trn*H-*psb*A (Appendix A). The NJ tree analysis of *trn*L-*trn*F showed that the experimental sample was located closest to *M. seguinii* from Vietnam, Cambodia and China (Appendix A). The experimental sample was finally authenticated as *M. seguinii* H.Lév on the basis of its morphological characteristics and DNA sequencing analysis. 

### 2.2. Isolation of Compounds from M. seguinii Using Molecular Networking

The 70% EtOH extract and its subfractions (25, 50, 75 and 100% MeOH eluent fractions) obtained via Diaion HP-20 column chromatography were analyzed by HPLC-qTOF MS/MS spectrometry. The GNPS web platform suggested 10 clusters (Appendix A) excluding the self-loop, and the precursor ion peak at *m*/*z* 449 matched as astilbin moiety according to the GNPS spectral library. The 100% MeOH-eluted fraction, which showed strong protective activity against cytotoxicity induced in A*β*_1-42_-transfected HT22 cells, was expected to contain the new targeted spectral nodes in the molecular network. Seven new flavanone glycoside derivatives (**1**–**7**), as well as five known compounds (**8**–**12**), were obtained from the 100% MeOH fraction through a series of chromatographic separations steps, including Diaion HP-20, Sephadex LH-20 and preparative HPLC (Figure 2).

### 2.3. Structural Determination of Compounds from M. seguinii

#### 2.3.1. (2*R*,3*R*)-4″-*O*-Galloylastilbin (**1**)

Compound **1** was isolated as a white amorphous powder with an ${\left[\alpha \right]}_{D}^{20}$ +55 (*c* 0.2, MeOH). Its molecular formula was suggested to be C_28_H_26_O_15_ on the basis of its deprotonated high-resolution electrospray ionization mass spectrometric (HRESIMS) ion peak at *m*/*z* 601.1221 [M − H]^−^ (calcd. for C_28_H_25_O_15_, 601.1193). The infrared (IR) spectrum showed the presence of hydroxyl and carboxyl bands at 3394 and 1640 cm^−1^, respectively. The ^1^H nuclear molecular resonance (NMR) data of **1** showed ABX aromatic proton signals at *δ*_H_ 6.99 (1H, d, *J* = 1.5 Hz), 6.87 (1H, dd, *J* = 8.1, 1.7 Hz) and 6.83 (1H, dd, *J* = 8.1 Hz), one tetra-substituted aromatic proton signal at *δ*_H_ 5.94 (1H, d, *J* = 1.8 Hz) and 5.91 (1H, d, *J* = 1.9 Hz) and two oxygenated methines at *δ*_H_ 5.08 (1H, d, *J* = 11.1 Hz) and 4.64 (1H, d, *J* = 11.1 Hz), which is suggestive of a flavanone moiety (Table 1). One rhamnose moiety at *δ*_H_ 4.06 (1H, s) and 1.09 (3H, d, *J* = 6.2 Hz) and one meta-coupled proton at *δ*_H_ 7.10 (2H, s) were observed in the ^1^H NMR data of **1**. The ^13^C NMR spectrum presented signals for 28 carbons, including two *α*,*β*-unsaturated carbonyl groups (*δ*_C_ 196.1 and 168.2), 18 aromatic carbon signals (*δ*_C_ 165.6–96.4), seven oxygenated carbons (*δ*_C_ 102.0–68.5) and one methyl carbon signal (*δ*_C_ 17.7) (Table 2). These results suggested that **1** was a 3,3′,4′,5,7-pentahydroxyflavanone with one rhamnose and one galloyl moiety. Compound **1** was similar to astilbin isolated from the leaves of *Engelhardia roxburghiana* [27], except for the attachment of one galloyl moiety to C-4′ of rhamnose in compound **1**. The analysis of HMBC and ^1^H-^1^H COSY showed that the anomeric proton of the rhamnose signal at *δ*_H_ 4.06 (1H, s) was correlated with C-3 (*δ*_C_ 78.6) and H-4″ of rhamnose at 5.04 (1H, t, *J* = 9.9 Hz), which was correlated with the galloyl ester carbon signal at *δ*_C_ 168.2 (Figure 3). The NMR coupling constants (*J*_2,3_) of C-2 and C-3 of **1** showed *J*_2,3_ = 11.1 Hz, which was suggestive of a *trans*-configuration and consequently, its relative configuration was [(2*R*,3*R*) or (2*S*,3*S*)] based on comparison with that of the reference [28]. The electronic circular dichroism (ECD) data of compound **1**, with a positive *n*→*π** (ca. 300–340 nm) and a negative *π*→*π** (ca. 290–300 nm), indicated a 2*R* configuration based on comparison with reported data (Appendix A) [29]. Thus, compound **1** was assigned as (2*R*,3*R*)-4″-*O*-galloylastilbin.

#### 2.3.2. (2*R*,3*S*)-4″-*O*-Galloylisoastilbin (**2**)

Compound **2** was obtained as a brownish gum with ${\left[\alpha \right]}_{D}^{20}$ −183 (*c* 0.2, MeOH). The molecular formula of C_28_H_26_O_15_ was determined from its negative HRESIMS ion peak at *m*/*z* 601.1186 [M − H]^−^ (calcd. for C_28_H_25_O_15_, 601.1193). The IR spectrum of **2** indicated characteristic bands of hydroxyl and carbonyl groups at 3393 and 1636 cm^−1^, respectively. The ^1^H and ^13^C NMR data (Table 1 and Table 2) were similar to those of compound **1** except for the presence of two oxygenated methines at *δ*_H_ 5.48 and 4.16 of H-2 and H-3 (d, *J* = 2.0 Hz), which indicated that the relative conformations of H-2 and H-3 of compound **2** were different from that of compound **1**. The NMR coupling constants (*J*_2,3_) and ECD spectra data (Appendix A) of compound **2** were indicative of a (2*R*,3*S*) conformation. Thus, compound **2** was assigned as (2*R*,3*S*)-4″-*O*-galloylisoastilbin.

#### 2.3.3. (2*R*,3*R*)-4″-*O*-(3‴,4‴-Dihydroxybenzoyl)astilbin (**3**)

Compound **3** was isolated as a brownish gum with ${\left[\alpha \right]}_{D}^{20}$ +3 (*c* 0.2, MeOH). The molecular formula of compound **3** was expected to be C_28_H_26_O_14_ according to the HRESIMS ion peak at *m*/*z* 585.1233 [M − H]^−^ (calcd. for C_28_H_25_O_14_, 585.1244), which was 16 amu less than that of compound **1** and was expected to result in the loss of a hydroxyl group. The IR spectrum indicated the presence of hydroxyl (3394 cm^−1^) and carbonyl (1640 cm^−1^) bands. On the basis of the ^1^H and ^13^C NMR data (Table 1 and Table 2), the aglycon and rhamnose moieties of compound **3** were similar to those of compound **1**, including the two oxygenated methines at *δ*_H_ 5.10 and 4.63 of H-2 and H-3 (d, *J* = 11.1 Hz). The ^1^H NMR and ^13^C NMR data of compound **3** showed ABX aromatic proton signals at *δ*_H_ 7.48 (1H, br s), 6.84 (2H, overlapped with H-5′) and 7.46 (1H, dd, *J* = 8.1, 2.0 Hz) and seven carbon signals (*δ*_C_ 122.6, 117.6, 146.6, 151.8, 115.9, 123.8 and 168.1), which indicated the presence of a 3,4-dihydroxybenzoic acid group. HMBC correlation showed that the anomeric proton of the rhamnose signal at *δ*_H_ 4.07 (1H, s) was correlated with C-3 at *δ*_C_ 78.7 and that H-4 of the rhamnose signal at 5.04 (1H, t, *J* = 9.8 Hz) was correlated with the 3,4-dihydroxybenzoic acid ester carbon signal at *δ*_C_ 168.1. The NMR spectrum and ECD spectra (Appendix A) of compound **3** indicated that the absolute configuration was (2*R*,3*R*). Thus, compound **3** was assigned as (2*R*,3*R*)-4″-*O*-(3‴,4‴-dihydroxybenzoyl)astilbin.

#### 2.3.4. (2*R*,3*R*)-4″-*O*-Vanilloylastilbin (**4**)

Compound **4**, obtained as a brownish gum with ${\left[\alpha \right]}_{D}^{20}$ −21 (*c* 0.2, MeOH), was deduced to have a molecular formula of C_29_H_28_O_14_ from its HRESIMS ion peak at *m*/*z* 599.1396 [M − H]^−^ (calcd. for C_29_H_27_O_14_, 599.1401). The IR spectrum showed the presence of hydroxyl (3414 cm^−1^) and carbonyl (1639 cm^−1^) bands. The ^1^H and ^13^C NMR data (Table 1 and Table 2) of compound **4** were similar to those of compound **3** except for one methoxy group at *δ*_H_ 3.90 (3H, s) and *δ*_C_ 56.5. HMBC correlations showed from the anomeric proton of the rhamnose signal at *δ*_H_ 4.08 (1H, s) to C-3 at *δ*_C_ 78.9, H-4 of the rhamnose signal at *δ*_H_ 5.07 (1H, t, *J* = 10.0 Hz) to the vanillic acid ester carbon signal at *δ*_C_ 167.9 and the methoxy group at 3.90 (3H, s) to C-3‴ (*δ*_C_ 148.7). The NMR spectrum and ECD spectra (Appendix A) of compound **4** indicated that the absolute configuration was (2*R*,3*R*). Hence, the structure of compound **4** was established as (2*R*,3*R*)-4″-*O*-vanilloylastilbin.

#### 2.3.5. (2*R*,3*S*)-4″-*O*-Vanilloylisoastilbin (**5**)

The molecular formula of compound **5**, obtained as a brownish gum with ${\left[\alpha \right]}_{D}^{20}$ −56 (*c* 0.2, MeOH), was deduced to be C_29_H_28_O_14_ from its HRESIMS ion peak at *m*/*z* 599.1382 [M − H]^−^ (calcd. for C_29_H_27_O_14_, 599.1401). The ^1^H and ^13^C NMR data (Table 1 and Table 2) were similar to those of compound **4** except for chemical shifts in two oxygenated methine signals at *δ*_H_ 5.47 (H-2) and 4.16 (H-3) and its coupling constants (d, *J* = 2.1 Hz). The ECD spectra of compound **5** indicated that the absolute configuration of **5** was (2*R*,3*S*) (Appendix A). Thus, compound **5** was characterized as (2*R*,3*S*)-4″-*O*-vanilloylisoastilbin.

#### 2.3.6. (*2R,3R*)-4″-O-(4‴-Hydroxybenzoyl)astilbin (**6**)

Compound **6** was obtained as a brownish gum with ${\left[\alpha \right]}_{D}^{20}$ −41 (*c* 0.2, MeOH). The molecular formula of **6** was inferred to be C_28_H_25_O_13_ based on its HRESIMS peak at *m*/*z* 569.1301 [M − H]^−^ (calcd. for C_28_H_25_O_13_, 569.1295). The molecular weight of compound **6**, which was 16 amu less than that of compound **3**, suggested the loss of a hydroxyl group. According to ^1^H and ^13^C NMR data (Table 1 and Table 2), A_2_B_2_ aromatic proton signals at *δ*_H_ 7.92 (2H, d, *J* = 8.7 Hz) and 6.85 (3H, overlapped with H-5′) and corresponding carbon signals at *δ*_C_ 122.3, 132.9, 116.2, 163.6 and 167.9 of **6** suggested the presence of a *para*-hydroxybenzoic acid group. The HMBC correlation between H-4″ at *δ*_H_ 5.07 and C-7‴ (*δ*_C_ 167.9) indicated the location of *p*-hydroxybenzoic acid. The absolute configuration of **6** was assigned as (2*R*,3*R*) based on the NMR spectrum and ECD spectra (Appendix A). Hence, compound **6** was identified as (2*R*,3*R*)-4″-*O*-(4‴-hydroxybenzoyl)astilbin.

#### 2.3.7. (2*R*,3*R*)-3″-*O*-E-Feruloylastilbin (**7**)

Compound **7** was isolated as a brownish gum with ${\left[\alpha \right]}_{D}^{20}$ +36 (*c* 0.2, MeOH). The molecular formula of compound **7** was deduced to be C_31_H_30_O_14_ from its HRESIMS ion peak at *m*/*z* 625.1554 [M − H]^−^ (calcd. for C_31_H_29_O_14_, 625.1557). On the basis of ^1^H and ^13^C NMR data (Table 1 and Table 2), the aglycon and rhamnose moieties of compound **7** were closely related to those of compound **1**. The existence of one ABX-type coupling system [*δ*_H_ 7.19 (1H, d, *J* = 1.8 Hz), 6.81 (1H, d, *J* = 8.0 Hz) and 7.08 (1H, dd, *J* = 8.2, 1.8 Hz)], a *trans*-*vinylic* group [*δ*_H_ 7.68 (1H, d, *J* = 15.9 Hz) and 6.41 (1H, d, *J* = 15.9 Hz)] and one oxygenated methyl proton at *δ*_H_ 3.90 (3H, s) indicated the presence of a *trans*-feruloyl moiety. In addition, the ^1^H-^1^H COSY spectrum and HMBC spectrum showed a correlation between H-3″ at *δ*_H_ 5.08 and C-9‴ at *δ*_C_ 168.9. These results indicated that the feruloyl moiety was linked to the C-3″ position of rhamnose (Figure 3). The NMR coupling constants (*J*_2,3_) and ECD spectra (Appendix A) resulted in the assignment of the absolute configuration of **7** as (2*R*,3*R*). The structure of compound **7** was elucidated as (2*R*,3*R*)-3″-*O*-*E*-feruloylastilbin. 

The five known compounds (**8**–**12**) were identified as neoastilbin (**8**), astilbin (**9**), isoastilbin (**10**), engeletin (**11**) and isoengeletin (**12**) by the comparison of the spectroscopic data with published data [28,30].

### 2.4. Neuroprotective Effects of Isolated Compounds (***1***–***12***)

The neuroprotective effects of all isolated compounds against the cytotoxicity induced by A*β*_42_ plasmids (transfected using Lipofectamine 2000 reagent) were measured in HT22 cells. To evaluate the cytotoxicity of the tested compounds, HT22 cells were incubated with the tested compounds at 20 μM. When cell viability was measured in a 3-(4,5-dimethylthiazol-2-yl)-2,5-diphenyl-2*H*-tetrazolium bromide (MTT) assay, no significant cytotoxicity was observed (Appendix A). After transfection for 10 h, compounds **2**, **6** and **7** showed neuroprotective effects against the cytotoxicity induced by A*β*_42_ plasmids at 20 μM (Figure 4A). As shown in Figure 4B, the protective effects of **2**, **6** and **7** showed dose-dependent activity (5, 10 and 20 μM) in the A*β*_42_-transfected HT22 cells. The intensity of green fluorescence was measured by pEGFP-C1/A*β*_1–42_ plasmid transfection in HT22 cells. Compounds **2**, **6** and **7** at 20 μM resulted in a moderately decreased fluorescence intensity compared with the results of vehicle treatment (Figure 5). 

## 3. Discussion

In this study, the studied plant was authenticated as *M. seguinii* H. Lév based on its morphological characteristics and DNA sequencing analysis. The misidentification of plants can result in toxic responses in patients who ingest the plants [24]. The expressed plant phenotype is described as a feature of the genotype that is modified in response to different environmental conditions [31]. Since the authentication of this plant was difficult, analyses of morphological characteristics, leaf and stem cross-sections, and the five DNA barcode sequences (*mat*K, *rbc*L, *trn*H-*psb*A, *trn*L-*trn*F and nrITS) were performed. Seven undescribed flavanonols (**1**–**7**) along with five known compounds (**8**–**12**) were isolated by MS/MS molecular networking from the stem of *M. seguinii*. Compounds **1**–**7** were 3,3′,4′,5,7-pentahydroxyflavanones with one rhamnose and a variety of attached benzoyl groups, including galloyl, 3,4-dihydroxybenzoyl, vanilloyl, 4-hydroxybenzoyl and *trans*-feruloyl groups. Flavanonol glycosides with aromatic groups are uncommon derivatives that can be isolated from plants. In particular, *M. seguinii* is unique in that flavanonol glycosides with various functional groups can be isolated from this single plant species. 

Compounds **2**, **6** and **7** showed significant neuroprotective activities against A*β*-induced cytotoxicity in A*β*_42_-transfected HT22 cells. The occurrence of AD is known to involve deposits of insoluble A*β* proteins and neurofibrillary tangles of tau-protein. Oxidative stress and inflammation are among the results of the production of these proteins [22,32]. Flavanonol glycosides from leaves of *Engelhardia roxburghiana* were found to have anti-inflammatory properties in an in vitro assay. These derivatives exert inhibitory effects on the mRNA expression of IL-1*β* and Cox-2 depending on the presence of galloyl or coumaroyl moieties [33]. The stereoisomers of astilbin based on the C-2 and C-3 configurations could affect the capacities of different antioxidants in the DPPH radical system and ABTS^+^ radical scavenging and anti-inflammatory effects by stimulating IL-1*β*, IL-6 and NO in RAW264.7 cells [34]. These results also suggest that the observed neuroprotective effects can be attributed to various aromatic groups, such as galloyl, hydroxybenzoyl and *trans*-feruloyl groups, and show different protective effects against A*β*-induced cytotoxicity depending on the type of substituent. Future research should focus on the structural specificity of the biological activity of phytochemicals. 

## 4. Materials and Methods

### 4.1. General Experimental Procedures

Optical rotations were measured on a JASCO P-2000 polarimeter (JASCO International Co. Ltd., Tokyo, Japan). IR data were collected using a Nicolet 6700 FT-IR spectrometer (Thermo Electron Corp., Waltham, MA, USA). ECD spectra were recorded using Chirascan Plus (Applied Photophysics Ltd., Surrey, UK). UHPLC systems (Ultimate 3000, Thermo Scientific, Milan, Italy) coupled to a Waters Xevo G2 QTOF MS spectrometer (Waters, Co., Milford, MA, USA) were used to generate HRESIMS values and perform LC-MS/MS analysis. Semipreparative HPLC was performed using a Gilson HPLC system with a UV/VIS-155 detector and a 321 pump. An RS Tech Optima Pak C18 column (10 × 250 mm, 10 μm) was used as the HPLC column. All solvents employed for extraction and isolation were of analytical grade. The 1D (^1^H and ^13^C) and 2D (HSQC, HMBC and ^1^H-^1^H COSY) NMR spectra were collected using JEOL 400 MHz (JEOL Ltd., Tokyo, Japan), Bruker Avance 500 MHz and Bruker Avance 800 MHz NMR spectrometers (Bruker, Billerica, MA, USA). Diaion^TM^ HP-20 ion exchange resin and GE Healthcare Sephadex^TM^ LH-20 (18–111 μm) were employed for column chromatography. Thin layer chromatography was performed with silica gel 60 F_254_ and RP-18 F_254_ plates.

### 4.2. Plant Material 

Dried stems of the studied plant material were collected in March 2017 from Gia Sinh Commune, Gia Vien District, Ninh Binh province, Vietnam (20°16′12.8″ N 105°51′59.7″ E). The sample was authenticated as *M. seguinii* H. Lév. based on morphological characteristics by Msc. Nghiem Duc Trong of the Department of Botany, Hanoi University of Pharmacy. The DNA sequencing of five markers, *mat*K, *rbc*L, *trn*H-*psb*A, *trn*L-*trn*F and nrITS, also supported for the species identity of the plant. A voucher specimen (HNIP-21-2017) was deposited in the Herbarium of the Hanoi University of Pharmacy, Hanoi, Vietnam. 

### 4.3. Morphology and DNA Sequencing Analysis of M. seguinii 

The sample was authenticated by matching its morphological features with available taxonomic descriptions of *M. seguinii* H. Lév. Photographs were obtained with a Canon EOS 60D camera and Canon 100 mm f2.8 IS Macro lens (Canon Inc. Tokyo, Japan). An EZ4 stereo microscope was used to analyze the features of *M. seguinii* leaf and stem cross-sections. Genomic DNA was extracted from a branch from which the outer skin was removed using a G-spinII Plant Genomic DNA extraction kit (Invitrogen, Seoul, Korea). The extracted genomic DNA was amplified by PCR using primers targeting four chloroplast genomic markers (*mat*K, *rbc*L, *trn*H-*psb*A and *trn*L-*trn*F) and the nuclear genome marker ITS region. The amplified PCRs products were purified using a PCR quick-spin^TM^ PCR Purification kit (Intron, Seongnam, Korea) and the DNA sequences were determined with the primers used for PCR. The DNA sequence was modified using Geneious 11.1.2 to determine the final DNA sequence. 

### 4.4. LC-MS/MS Analysis and Molecular Networking 

LC-MS/MS analysis was carried out by using UHPLC systems (Ultimate 3000, Thermo Scientific, Milan, Italy) coupled to a Waters Xevo G2 QTOF MS spectrometer (Waters, Co., Milford, MA, USA) with an Acquity UPCL BEH C18 column (2.1 × 100 mm, 1.7 μm). The mobile phase was composed of 0.1% formic acid in water (A) and 0.1% formic acid in acetonitrile (B). The gradient program was as follows: 0–20 min, linear gradient of 10% to 90% B; 20–22 min, isocratic at 100% B; 22–24 min, return to the initial conditions. The flow rate was 0.3 mL/min, and the injection volume was 2 μL during the acquisition of negative polarity. The MS/MS analysis was performed in data-dependent scan mode, ranging from *m*/*z* = 100 to *m*/*z* 1500. All LC-MS/MS data were converted to mzXML files by MZmine-2.38. Molecular networking was carried out via the GNPS web platform (https://gnps.ucsd.edu, accessed on 7 January 2019). The spectral networks were visualized in Cytoscape 3.7.1 (www.cytoscape.org, accessed on 8 January 2019). 

### 4.5. Extraction and Isolation of Flavonoids from M. seguinii

Dried stems of *M. seguinii* (1.5 kg) were extracted with 70% ethanol under ultrasonication, which was conducted 3 times over 3 days. The extract was evaporated under vacuum to yield the extract (150 g). The 70% ethanol extract (150 g) was subjected to chromatography on a Diaion HP-20 column, with elution via a H_2_O/MeOH gradient system (25:75→50:50→25:75→0:100) to obtain four fractions (fractions 1–4). Fraction 4 (6 g) was subjected to chromatography on a BUCHI Reveleris^®^ purification system with an RP-18 column (C18, 40 μm, 120 g), with elution via a H_2_O/MeOH gradient (80:20→0:100, *v*:*v*, run time: 200 min) to obtain 14 fractions (fractions 4.1–4.14). Fraction 4.5 (500 mg) was subjected to Sephadex LH-20 chromatography with MeOH to yield five fractions (5.5.1–4.5.5). Compounds **1** and **2** from fraction 4.5.4 were purified on a Gilson HPLC system with a CH_3_CN/H_2_O isocratic solvent (26:74, *v*:*v*) and a run time of 50 min (com. **1**: 32 min, 4.9 mg and com. **2**: 35 min, 2.8 mg). Fraction 4.7 (460 mg) was also subjected to chromatography by using a Sephadex LH-20 (2.5 × 70 cm) system with MeOH to obtain seven fractions (4.7.1–4.7.7). Compound **3** from fraction 4.7.5 was isolated on a Gilson HPLC system with a CH_3_CN/H_2_O isocratic solvent (27:73, *v*:*v*) and a run time of 50 min (com. **3**: 35 min, 3.9 mg). Compound **7** from fraction 4.7.4 was analyzed on the Gilson HPLC system with a CH_3_CN/H_2_O isocratic solvent (33:67, *v*:*v*) and a run time of 50 min (com. **7**: 42 min). Fraction 4.8 was subjected to chromatography on a Sephadex LH-20 system (2.5 × 70 cm) by using MeOH to obtain seven fractions (4.8.1–4.8.7). Compounds **4** and **6** from fraction 4.8.4 (130 mg) were analyzed on a Gilson HPLC system with a CH_3_CN/H_2_O isocratic solvent (27:73, *v*:*v*) and a run time of 50 min (com. **4**: 45 min, 3.5 mg and com. **6**: 40 min, 3.0 mg). Fraction 4.6 was purified on a Sephadex LH-20 system (2.5 × 70 cm), with elution by MeOH to obtain five fractions (4.6.1–4.6.5). Compound **5** from fraction 4.6.5 was carried out on a Gilson HPLC system with a CH_3_CN/H_2_O isocratic solvent (30:70, *v*:*v*) and a run time of 50 min (com. **5**: 42 min, 2.0 mg). Compounds **8**, **9** and **10** were purified from fraction 4.3 using a Gilson HPLC system with a CH_3_CN/H_2_O isocratic solvent (20:80, *v*:*v*) and a run time of 45 min (com. **8**: 26 min, 3.9 mg, com. **9**: 30 min, 14.6 mg and com. **10**: 42 min, 3.3 mg). From fraction 4.4, compounds **11** and **12** were isolated by using a Gilson HPLC system with a CH_3_CN/H_2_O isocratic solvent (25:75, *v*:*v*) and a run time of 30 min (com. **11**: 21 min, 3.0 mg and com. **12**: 28 min, 2.4 mg). The HPLC isolation conditions were as follows: flow rate 2 mL/min, UV detection at 210 and 254 nm, and a Optimapack C18 column (10 × 250 mm, 5 μm). 

### 4.6. Spectroscopic and Physical Characteristic of Compounds

*(2R,3R)-4″-O-galloylastilbin* (**1**): Amorphous powder; ${\left[\alpha \right]}_{D}^{20}$ +55 (*c* 0.2, MeOH); UV (MeOH) *λ*_max_ (log *ε*) 218 (3.2), 289 (3.0) nm; IR *ν*_max_ 3394, 2980, 1640, 1524, 1454, 1240, 1032 cm^−1^; ECD (MeOH) *λ*_max_ (Δ*ε*) 224 (7.4), 257 (1.6), 296 (–10.8), 331 (2.9) nm; ^1^H and ^13^C NMR, see Table 1 and Table 2; HRESIMS *m*/*z* 601.1221 [M – H]^−^ (calcd. for C_28_H_25_O_15_, 601.1193).

*(2R,3S)-4″-O-galloylisoastilbin* (**2**): Brownish gum; ${\left[\alpha \right]}_{D}^{20}$ −183 (*c* 0.2, MeOH); UV (MeOH) *λ*_max_ (log *ε*) 200 (3.1), 216 (3.0), 291 (2.7) nm; IR *ν*_max_ 3393, 2980, 1636, 1516, 1448, 1229, 1032 cm^−1^; ECD (MeOH) *λ*_max_ (Δ*ε*) 224 (−8.6), 251 (−0.2), 283 (−4.4), 310 (−3.9), 343 (3.5) nm; ^1^H and ^13^C NMR, see Table 1 and Table 2; HRESIMS *m*/*z* 601.1186 [M − H]^−^ (calcd. for C_28_H_25_O_15_, 601.1193).

*(2R,3R)-4″-O-(3‴,4‴-dihydroxybenzoyl)astilbin* (**3**): Brownish gum; ${\left[\alpha \right]}_{D}^{20}$ +3 (*c* 0.2, MeOH); UV (MeOH) *λ*_max_ (log *ε*) 201 (3.1), 218 (2.9), 270 (2.4), 292 (2.6) nm; IR *ν*_max_ 3394, 2980, 1641, 1520, 1448, 1289, 1089, 1032 cm^−1^; ECD (MeOH) *λ*_max_ (Δ*ε*) 225 (7.6), 257 (1.4), 297 (−10.4), 330 (2.9) nm; ^1^H and ^13^C NMR, see Table 1 and Table 2; HRESIMS *m*/*z* 585.1233 [M − H]^−^ (calcd. for C_28_H_25_O_14_, 585.1244).

*(2R,3R)-4″-O-vanilloylastilbin* (**4**): Brownish gum; ${\left[\alpha \right]}_{D}^{20}$ −21 (*c* 0.2, MeOH); UV (MeOH) *λ*_max_ (log *ε*) 201 (2.8), 218 (2.7), 268 (2.2), 292 (2.4) nm; IR *ν*_max_ 3414, 2938, 1639, 1515, 1453, 1281, 1114, 1032 cm^−1^; ECD (MeOH) *λ*_max_ (Δ*ε*) 226 (9.0), 257 (1.9), 297 (−12.0), 332 (3.5) nm; ^1^H and ^13^C NMR, see Table 1 and Table 2; HRESIMS *m*/*z* 599.1396 [M − H]^−^ (calcd. for C_29_H_27_O_14_, 599.1401).

*(2R,3S)-4″-O-vanilloylisoastilbin* (**5**): Brownish gum; ${\left[\alpha \right]}_{D}^{20}$ −56 (*c* 0.2, MeOH); UV (MeOH) *λ*_max_ (log *ε*) 200 (3.2), 218 (3.1), 270 (2.7), 294 (2.9) nm; IR *ν*_max_ 3394, 2980, 1637, 1515, 1455, 1285, 1032 cm^−1^; ECD (MeOH) *λ*_max_ (Δ*ε*) 227 (−9.4), 253 (−0.2), 284 (−4.4), 310 (−3.9), 344 (5.3) nm; ^1^H and ^13^C NMR, see Table 1 and Table 2; HRESIMS *m*/*z* 599.1382 [M − H]^−^ (calcd. for C_29_H_27_O_14_, 599.1401).

*(2R,3R)-4″-O-(4‴-hydroxybenzoyl)astilbin* (**6**): Brownish gum; ${\left[\alpha \right]}_{D}^{20}$ −42 (*c* 0.2, MeOH); UV (MeOH) *λ*_max_ (log *ε*) 200 (2.8), 216 (2.6), 261 (2.2), 289 (2.3) nm; IR *ν*_max_ 3370, 2980, 1640, 1514, 1452, 1275, 1165, 1032 cm^−1^; ECD (MeOH) *λ*_max_ (Δ*ε*) 226 (7.7), 258 (1.3), 295 (−8.8), 332 (2.4) nm; ^1^H and ^13^C NMR, see Table 1 and Table 2; HRESIMS *m*/*z* 569.1301 [M − H]^−^ (calcd. for C_28_H_25_O_13_, 569.1295).

*(2R,3R)-3″-O-E-feruloylastilbin* (**7**): Brownish gum; ${\left[\alpha \right]}_{D}^{20}$ +36 (*c* 0.2, MeOH); UV (MeOH) *λ*_max_ (log *ε*) 200 (2.9), 218 (2.8), 230 (2.7), 294 (2.7), 324 (2.6) nm; IR *ν*_max_ 3340, 2972, 1638, 1515, 1270, 1162, 1032 cm^−1^; ECD (MeOH) *λ*_max_ (Δ*ε*) 224 (13.0), 251 (2.1), 294 (−6.9), 326 (5.4) nm; ^1^H and ^13^C NMR, see Table 1 and Table 2; HRESIMS *m*/*z* 625.1554 [M − H]^−^ (calcd. for C_31_H_29_O_14_, 625.1557).

### 4.7. Cell Culture and Cell Viability Assay

HT22 immortalized mouse hippocampal neuronal cells at 70% confluence were cultured in DMEM (HyClone, Logan, UT, USA) supplemented with 10% fetal bovine serum (FBS), 100 U/mL penicillin and 100 μg/mL streptomycin for 48 h. Then, the HT22 cells were seeded in 96-well plates at 3000 cells/well. After incubation for 24 h, the test compounds (20 μM) were added to the cells for 24 h and cell viability was measured via the MTT (Sigma, St Louis, MO, USA) reagent method. A 20 μL aliquot of MTT solution (2 mg/mL) was injected into each well, and the plate was incubated for 3 h in the dark. After the resultant formazan crystals were dissolved in DMSO, the absorbance was measured with a microplate reader (VersaMaxTM, Randor, PA, USA) at 570 nm.

### 4.8. Cytotoxicity Assay of Aβ_1–42_-Transfected HT22 Cells

HT22 cells were seeded in 96-well plates at 3000 cells/well and incubated at 37 °C with 5% CO_2_ for 3 h [35,36]. The cells in each well were transfected with 0.2 μg of the pEGFP-C1/A*β*_42_ plasmid (originating from Professor Junsoo Park, Yonsei University, Seoul, Korea) with Lipofectamine 2000 reagent (Invitrogen, Carlsbad, CA, USA). After transfection for 10 h, the tested compounds dissolved in the medium were added to the seeded cells, and the cells were incubated for 24 h. Then, 20 μL of MTT solution (2 mg/mL) was added, and the cells were incubated for 3 h in the dark. The cells were subsequently washed with phosphate-buffered saline (PBS) (Takara, Kusatsu, Japan), and 100 mL of DMSO was added to solubilize formazan. The absorbance at 570 nm was measured with a microplate reader (VersaMaxTM, Randor, PA, USA). Fluorescence imaging was conducted by using a fluorescence microscope (Olympus ix70, Tokyo, Japan).

### 4.9. Statistical Analysis

Data were evaluated as the mean ± SD of three independent experiments. Data were processed by analysis of variance (ANOVA) which was conducted using SPSS Statistics 23 (SPSS, Inc., Chicago, IL, USA). Statistically significant *p* values were established at * *p* < 0.05, ** *p* < 0.01, *** *p* < 0.001.

## 5. Conclusions

The sample studied herein was authenticated as *M. seguinii* H. Lév on the basis of its morphological features and the DNA barcode technique. Seven new flavanonol glycosides with various conjugated aromatic groups and five known flavanonol glycosides were isolated by LC-MS/MS molecular networking, and their neuroprotective effects against A*β*_42_-induced cytotoxicity were measured. Among the isolated compounds, compounds **2**, **6** and **7** showed potential neuroprotective activity. Compound **2**, with a galloyl group and a *cis*-configuration at the C-2 and C-3 positions, showed the strongest protective activity. These results suggested that flavanonol glycosides from *M. seguinii* could be good candidates for use in AD treatment.

## Figures and Tables

**Figure 1 pharmaceuticals-14-00911-f001:**
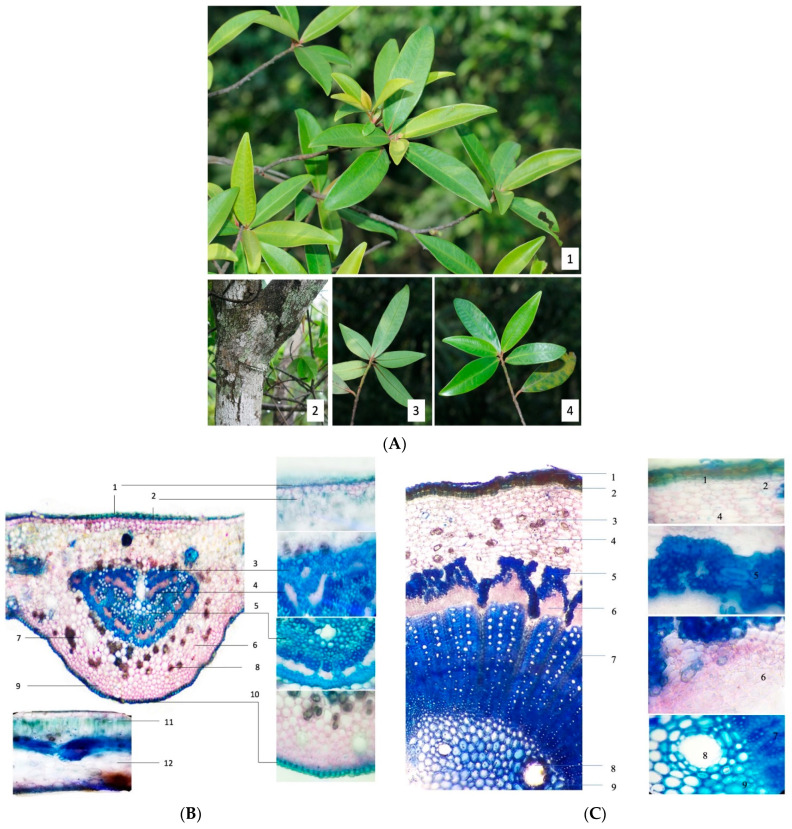
(**A**) Morphological characteristics of *Myrsine seguinii* H. Lév. (1) branchlets, (2) plant bark, (3) abaxial leaf, (4) adaxial leaf; (**B**) Cross-section of the leaf of *M. seguinii*. (1) upper epidermis, (2) upper collenchyma, (3) sclerencyma, (4) phloem, (5) xylem, (6) collenchymal tissues, (7) schizogenous cavity, (8) calcium oxalate crystals, (9) lower chollenchyma, (10) lower epidermis, (11) palisade mesophyll, (12) spongy mesophyll; (**C**) Cross-section of the stem of *M. seguinii*. (1) cork, (2) parenchyma cells, (3) calcium oxalate crystals, (4) parenchyma, (5) sclerenchyma cells, (6) secondary phloem, (7) secondary xylem, (8) schizogenous cavities, (9) primary xylem.

**Figure 2 pharmaceuticals-14-00911-f002:**
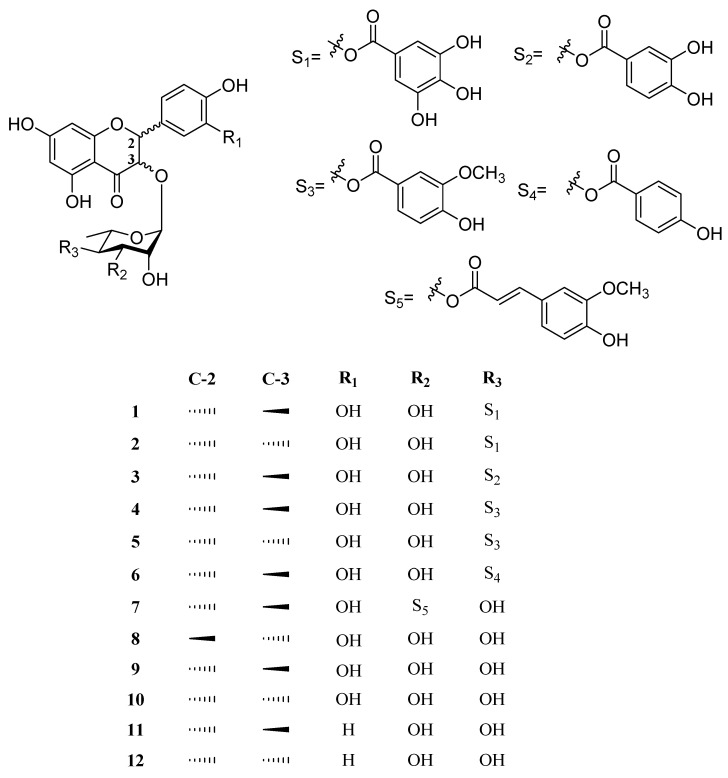
The structures of the isolated flavanonol glycosides from *M. seguinii*.

**Figure 3 pharmaceuticals-14-00911-f003:**
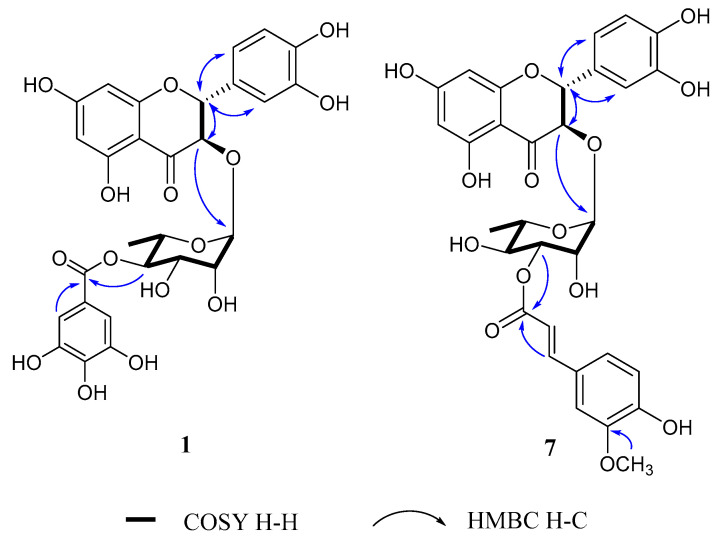
The key COSY and HMBC correlations of compounds **1** and **7**.

**Figure 4 pharmaceuticals-14-00911-f004:**
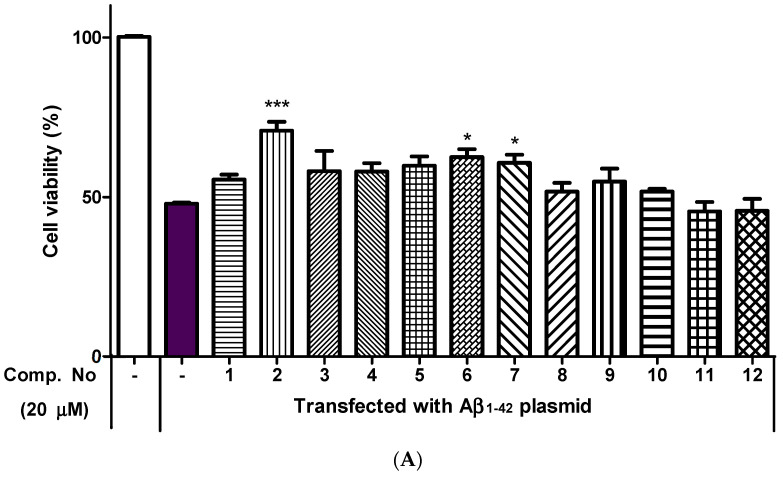
(**A**) Neuroprotective effects of compounds **1**–**12** against cytotoxicity induced by A*β*_42_ plasmid transfection into HT22 cells. The cells were transfected with A*β*_42_ plasmids for 10 h and treated with the test compounds at 20 μM. (**B**) Neuroprotective effects of compounds **2**, **6** and **7** at different concentrations. After incubation for 24 h, cell viability was measured using the MTT assay. Each value was calculated as the mean ± SD (*n* = 3); * *p* < 0.05, ** *p* < 0.01, *** *p* < 0.001 compared with the untreated A*β*_42_ transfection group.

**Figure 5 pharmaceuticals-14-00911-f005:**
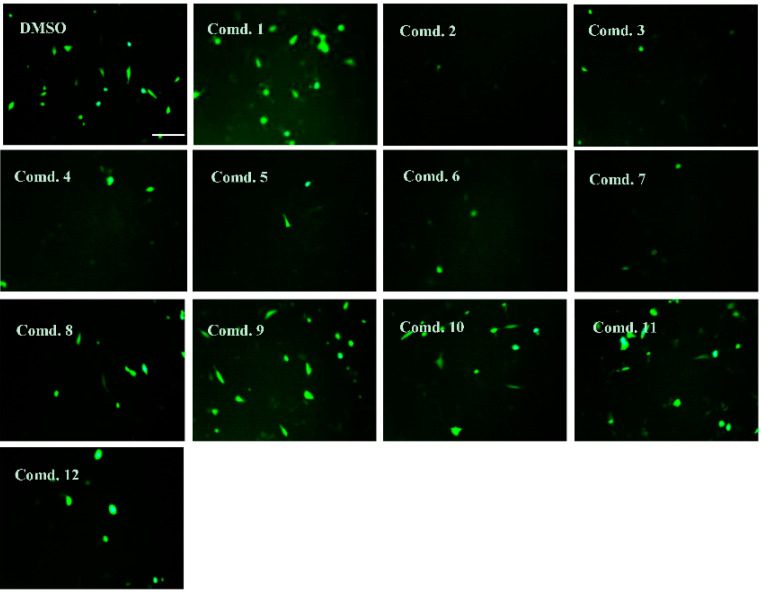
Effects of compounds **1**–**12** on the intensity of the green fluorescence induced by pEGFP-C1/A*β*_42_ plasmid transfection using Lipofectamine in HT22 cells after 10 h of transfection. The transfected cells were continually exposed to the test compounds at 20 μM. After incubation for 24 h, the cells were visualized using fluorescence microscopy.

**Table 1 pharmaceuticals-14-00911-t001:** ^1^H NMR data for compounds **1**–**7** in CD_3_OD (*δ*_H_ in ppm, mult. *J* in Hz).

No.	1 ^a^	2 ^a^	3 ^a^	4 ^b^	5 ^a^	6 ^b^	7 ^c^
2	5.08, d (11.1)	5.48, d (2.0)	5.10, d (11.2)	5.09, d (11.0)	5.47, d (2.1)	5.09, d (11.1)	5.12, d (11.0)
3	4.64, d (11.1)	4.16, d (2.2)	4.63, d (11.1)	4.63, d (11.0)	4.16, d (2.2)	4.63, d (11.1)	4.61, d (11.0)
4							
5							
6	5.94, d (1.8)	6.00, d (1.9)	5.93, d (1.7)	5.92, d (1.7)	6.00, d (2.1)	5.94, d (2.0)	5.93, d (2.2)
7							
8	5.91, d (1.9)	5.93, d (1.9)	5.91, d (1.7)	5.90, d (1.6)	5.94, d (2.1)	5.92, d (1.9)	5.91, d (2.1)
9							
10							
1′							
2′	6.87, dd (8.1, 1.7)	6.85, dd (8.3, 1.8)	6.99, d (1.6)	6.98, d (1.3)	6.87, br s	6.99, d (1.8)	6.98, d (2.0)
3′	6.83, d (8.1)	6.89, d (8.1)					
4′							
5′			6.84, overlap	6.82, d (8.1)	6.87, br s	6.85, overlap	6.80, d (8.0)
6′	6.99, d (1.5)	7.08, d (1.6)	6.87, d (8.1, 1.7)	6.87, overlap	7.03, br s	6.87, dd (8.1, 1.8)	6.87, dd (8.1, 2.0)
1″	4.06, s	5.02, br s	4.07, br s	4.08, s	5.02, d (1.2)	4.08, br s	4.03, d (1.3)
2″	3.59, br s	3.72, m	3.60, m	3.59, m	3.74, dd (3.3, 1.5)	3.60, m	3.76, dd (3.1, 1.8)
3″	3.93, dd (9.8, 1.3)	3.68, dd (9.7, 3.3)	3.94, dd (9.8, 3.2)	3.95, dd (9.8, 3.1)	3.70, dd (9.8, 3.4)	3.94, dd (9.8, 3.2)	5.08, dd (10.0, 3.2)
4″	5.04, t (9.9)	4.85, m	5.04, t (9.8)	5.07, t (10.0)	4.95, t (9.9)	5.07, t (9.9)	3.59, m
5″	4.59, d (9.9, 6.3)	2.43, dq (12.4, 6.1)	4.58, d (9.9, 6.3)	4.56, dq (12.4, 6.1)	2.43, dd (9.9, 6.2)	4.58, dd (9.9, 6.2)	4.45, dd (9.7, 6.2)
6″	1.09, d (6.2)	0.81, d (6.2)	1.10, d (6.3)	1.09, d (6.2)	0.82, d (6.3)	1.09, d (6.2)	1.24, d (6.2)
1‴							
2‴	7.10, s	7.11, s	7.48, br s	7.57, d (1.5)	7.55, d (1.9)	7.92, d (8.7)	7.19, d (1.8)
3‴						6.85, overlap	
4‴							
5‴			6.84, overlap	6.87, overlap	6.96, d (8.4)	6.85, overlap	6.81, d (8.0)
6‴	7.10, s	7.11, s	7.46, dd (8.1, 2.0)	7.60, dd (8.3, 1.7)	7.66, dd (8.3, 2.0)	7.92, d (8.7)	7.08, dd (8.2, 1.8)
7‴							7.68, d (15.9)
8‴							6.41, d (15.9)
9‴							
OCH_3_				3.90, s	3.96, s		3.90, s

^a^ measured at 400 MHz. ^b^ measured at 500 MHz. ^c^ measured at 800 MHz.

**Table 2 pharmaceuticals-14-00911-t002:** ^13^C NMR data for compounds **1**–**7** in CD_3_OD (*δ*_C_ in ppm).

No.	1 ^a^	2 ^a^	3 ^a^	4 ^b^	5 ^a^	6 ^b^	7 ^c^
2	84.0	82.0	84.0	84.0	82.0	84.0	83.9
3	78.6	74.7	78.7	78.9	74.7	78.8	78.9
4	196.1	193.6	196.1	196.1	193.8	196.1	195.8
5	165.6	164.6	165.6	164.2	164.2	165.6	165.6
6	97.5	96.4	97.5	97.5	97.5	97.5	97.5
7	168.9	169.5	168.9	168.8	169.2	168.8	168.8
8	96.4	97.6	96.3	96.3	96.3	96.3	96.3
9	164.2	166.4	164.2	165.6	166.3	164.2	164.2
10	102.4	101.6	102.5	102.5	101.6	102.5	102.5
1′	129.2	129.2	129.2	129.2	128.8	129.2	129.2
2′	120.5	118.9	115.5	115.5	118.9	115.6	115.4
3′	116.3	116.2	146.1	146.6	146.6	146.6	146.6
4′	147.5	146.6	147.5	147.5	146.6	147.5	147.4
5′	146.6	146.3	116.3	116.3	116.1	116.4	116.3
6′	115.5	115.2	120.5	120.5	114.9	120.6	120.5
1″	102.0	99.2	102.1	102.2	99.4	102.2	102.1
2″	71.9	72.1	71.9	71.9	72.2	71.9	69.7
3″	70.4	70.3	70.4	70.3	70.2	70.4	75.4
4″	75.5	74.9	75.5	75.7	74.9	75.5	71.3
5″	68.5	67.9	68.4	68.6	68.0	68.6	70.7
6″	17.7	17.7	17.8	17.8	17.7	17.8	17.9
1‴	121.5	121.4	122.6	122.6	122.4	122.3	127.8
2‴	110.2	110.6	117.6	113.7	113.8	132.9	111.7
3‴	146.5	146.3	146.6	148.7	148.7	116.2	149.3
4‴	139.8	139.8	151.8	152.9	152.9	163.6	150.6
5‴	146.5	146.3	115.9	115.9	116.2	116.2	116.5
6‴	110.2	110.6	123.8	125.3	125.9	132.9	124.1
7‴	168.2	168.1	168.1	167.9	168.0	167.9	146.9
8‴							115.8
9‴							168.9
OCH_3_				56.5	56.7		56.4

^a^ measured at 100 MHz. ^b^ measured at 125 MHz. ^c^ measured at 200 MHz.

## Data Availability

The data presented in this study are available in article and Appendix A.

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
