# Peer review of "Flavanonol Glycosides from the Stems of Myrsine seguinii and Their Neuroprotective Activities"

_pharmaceuticals, 2021, doi:10.3390/ph14090911_

Round 1
Reviewer 1 Report
In this Manuscript, the authors investigate the active compounds in ethanolic extract of M. seguinii plant, which can be used in the treatment of people suffering from Alzheimer's disease. The experiments are well planned and described in detail. Detailed Supplementary Material deserves attention.
However, some things in my opinion need to be improved:
Please improve the quality of the Fig. 4
line 138 "matched an astilbin" should be "matched as"
line 143 "MeOH fraction via a series" I think it would be better "through a series"
Line 356 "(10 x 250mm, 10 um particle size, Korea)" please remove "partice size" in in standard chromatographic nomenclature this value always represents the particle size, also "Korea" should not be placed here
line 390 please rewrite the specification of the chromatographic column in the similar way as in 356 line
Line 402 "extracted in 70% ethanol" I think it should be "extracted with", we can dissolve something in something but extract rather with
Caption to Fig S1 in Supplementary Materials "DAN sequences" I think it should be "DNA sequences"
Good luck!
Author Response
In this Manuscript, the authors investigate the active compounds in ethanolic extract of M. seguinii plant, which can be used in the treatment of people suffering from Alzheimer's disease. The experiments are well planned and described in detail. Detailed Supplementary Material deserves attention. However, some things in my opinion need to be improved:
Please improve the quality of the Fig. 4
→ Answer: Thank you for the editor’s comments. We improved the quality of the figure 4 in the revised manuscript.
line 138 "matched an astilbin" should be "matched as"
→ Answer: We corrected the sentence based on editor’s comments in the revised manuscript.
line 143 "MeOH fraction via a series" I think it would be better "through a series"
→ Answer: We changed the sentence based on editor’s comments in the revised manuscript.
Line 356 "(10 x 250mm, 10 um particle size, Korea)" please remove "partice size" in in standard chromatographic nomenclature this value always represents the particle size, also "Korea" should not be placed here
→ Answer: We removed the “particle size” and “Korea” based on editor’s comments in the revised manuscript.
line 390 please rewrite the specification of the chromatographic column in the similar way as in 356 line
→ Answer: We changed the sentence based on editor’s comments in the revised manuscript.
Line 402 "extracted in 70% ethanol" I think it should be "extracted with", we can dissolve something in something but extract rather with
→ Answer: We changed the sentence based on editor’s comments in the revised manuscript.
Caption to Fig S1 in Supplementary Materials "DAN sequences" I think it should be "DNA sequences"
→ Answer: These mistakes have been corrected in the revised manuscript.
Reviewer 2 Report
This work has a certain novelty , but may not be suitable for publication of Pharmaceuticals and the main issues are as follows :
- Can you provide the infrared spectra of these compounds
- Can you provide the melting point data of compound 1 and the yields of all compounds .
- According to your statements , Myrsine seguinii has traditionally been used to treat inflammation , why not evaluate the anti-inflammatory activity of these compounds but instead evaluate their anti-Alzheimer's effect ?
- Be there any research on the components of Myrsine seguinii in neurodegenerative diseases ?
- Can you explain how to determine the final dose concentration?
- According to your exposition , have you compared the protective activity against cytotoxicity of 100% MeOH-eluted fraction and each compound at the same dose ?
- Can some animal experiments be used to evaluate related activities in vivo and set up a positive control group to make the experiment more complete ?
- Can you analyze the structure-activity relationship of this type of compound ?
- Please carefully check the format of the reference and some spelling errors , such as " dot " on line 584 .
Author Response
This work has a certain novelty, but may not be suitable for publication of Pharmaceuticals and the main issues are as follows:
- Can you provide the infrared spectra of these compounds
→ Answer: Thank you for the reviewer’s comments. Even if we have already added all IR data from lines 425 to 459 of part “4.6. Spectroscopic and Physical Characteristic of Compounds”, we newly provided all original IR spectra of compounds 1-7 in revised supplementary materials. Editor can see our added IR spectra at Figures S4, S11, S18, S25, S32, S39, and S46 in revised supplementary file.
- Can you provide the melting point data of compound 1 and the yields of all compounds
→ Answer: We provided the yields of all compounds in 4.5. Extraction and isolation of flavonoids from M. seguinii section in the revised manuscript. We were unable to measure the melting point data due to sample loss and instrument failure. Recently, most of natural product journals do not need necessarily require the measurement of the melting point by adding all spectroscopic data at supplementary data file. Thus, we would like to understand that it cannot be measured by sample loss and instrument problems.
- According to your statements, Myrsine seguinii has traditionally been used to treat inflammation, why not evaluate the anti-inflammatory activity of these compounds but instead evaluate their anti-Alzheimer's effect?
→ Answer: It was possible to predict the expected structures through molecular networking and HRESIMS chemical formula, and some of their derivatives have already been reported for anti-inflammatory activities (reference 33).
- Xin, W.; Huang, H.; Yu, L.; Shi, H.; Sheng, Y.; Wang, T.T.Y.; Yu, L. Three new flavanonol glycosides from leaves of Engelhardia roxburghiana, and their anti-inflammation, antiproliferative and antioxidant properties. Food Chem. 2012, 132, 788−798, doi:10.1016/j.foodchem.2011.11.038.
The accumulation of Aβ peptides in brain of the Alzheimer disease (AD) is an important trigger for microglial activation. When AD is persistent and worsening, microglial cells are chronically activated, resulting in the production of a variety of pro-inflammatory cytokines and neurotoxic compounds, including interleukin-1β (IL-1β), IL-6, tumour necrosis factor-alpha (TNF-α), nitric oxide (NO), and reactive oxygen species (ROS). Many researchers have been reported that neuroinflammation in AD by these inflammatory mediators have injurious effects on neurodegenerative diseases and may result in brain damage. Therefore, we evaluated their anti-Alzheimer’s effect for novelty of these compounds.
- Be there any research on the components of Myrsine seguinii in neurodegenerative diseases?
→ Answer: Until recently, there were no studies on neurodegenerative diseases of the components of Mysine sequinii such as prenylated benzoic acids and hydroquinones. However, there have been many studies on flavonoids, several flavonoids (qercetin glycosides, myricetin glycosides…) have reported positive effects against neurodegenerative diseases (dementia and Alzheimer’s disease).
- Can you explain how to determine the final dose concentration?
→ Answer: We conducted the cytotoxicity assay of compounds at concentration 20 μM that there was no change in cell viability and the morphology of the cells. We determined the final dose concentration according to results of cytotoxicity assay.
- According to your exposition, have you compared the protective activity against cytotoxicity of 100% MeOH-eluted fraction and each compound at the same dose ?
→ Answer: Cytotoxicity experiments were commonly conducted at a concentration of 10 μg/mL for fractions and 20 μM for single compound, respectively.
- Can some animal experiments be used to evaluate related activities in vivo and set up a positive control group to make the experiment more complete ?
→ Answer: As a follow-up study, we plan to study more biological activities by conducting animal experiments.
- Can you analyze the structure-activity relationship of this type of compound ?
→ Answer: If we had isolated a more stereoisomers of C-2 and 3 position of this type of compounds, we could analyze more interesting results on the structure-activity relationship, and we could infer the results from the content mentioned in the discussion section.
- Please carefully check the format of the reference and some spelling errors, such as " dot " on line 584 .
→ Answer: The mistakes have been corrected in the revised manuscript.
Round 2
Reviewer 2 Report
I think this study need to add some experiments needed .